# Design of a Smart Factory Based on Cyber-Physical Systems and Internet of Things towards Industry 4.0

**Mutaz Ryalat *** ⓘ**, Hisham ElMoaqet** ⓘ **and Marwa AlFaouri**

Department of Mechatronics Engineering, German Jordanian University, Amman 11180, Jordan
* Correspondence: mutaz.ryalat@gju.edu.jo

**Abstract:** The rise of Industry 4.0, which employs emerging powerful and intelligent technologies and represents the digital transformation of manufacturing, has a significant impact on society, industry, and other production sectors. The industrial scene is witnessing ever-increasing pressure to improve its agility and versatility to accommodate the highly modularized, customized, and dynamic demands of production. One of the key concepts within Industry 4.0 is the smart factory, which represents a manufacturing/production system with interconnected processes and operations via cyber-physical systems, the Internet of Things, and state-of-the-art digital technologies. This paper outlines the design of a smart cyber-physical system that complies with the innovative smart factory framework for Industry 4.0 and implements the core industrial, computing, information, and communication technologies of the smart factory. It discusses how to combine the key components (pillars) of a smart factory to create an intelligent manufacturing system. As a demonstration of a simplified smart factory model, a smart manufacturing case study with a drilling process is implemented, and the feasibility of the proposed method is demonstrated and verified with experiments.

**Keywords:** cyber-physical systems (CPS); Industry 4.0; Internet of Things (IoT); robotics; smart factory

## 1. Introduction

As today's markets are being reshaped by groundbreaking technologies, there is more pressure on the industry to become more flexible and adaptable in order to meet the changing needs of markets. With more competition in efficiency, productivity, and quality in the global market, companies need to make big changes to their production plans, technologies, and management [1]. Thanks to technological advancements, many different industries have been able to boost their performance and productivity through the use of automation, digitalization, and artificial intelligence, as well as the unprecedented availability and affordability of computing power, smart sensors, data acquisition systems, intelligent robotics, information and communication technology (ICT), the introduction of the Internet of Things (IoT), Big Data, and cloud computing [2].

The transition from classical manufacturing to intelligent manufacturing, which is the natural consequence of digitalization and new advanced technologies, has given rise to a fourth industrial revolution, or Industry 4.0. The term was first coined in Germany in 2011 [3], and it comprehends new manufacturing techniques, ones that incorporate digitalization into the industrial world through the use of IoT. The success of Industry 4.0 relies heavily on the smart deployment of key enabling technologies (KETs) [4], being the pillars of this revolution. Figure 1 depicts the most common Industry 4.0 pillars identified in the literature.

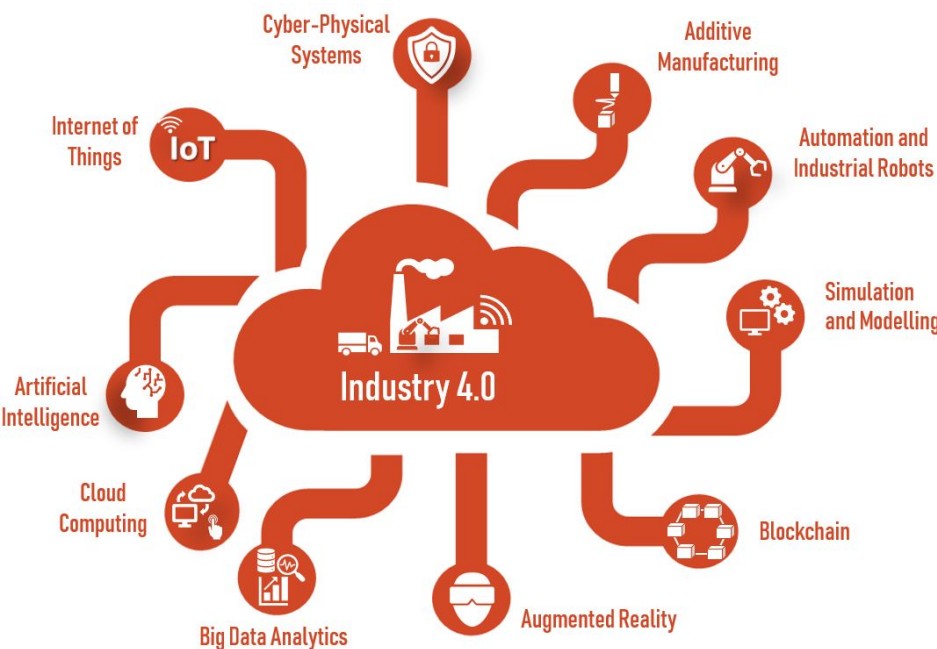

**Figure 1.** Pillars of Industry 4.0.

The following subsections will highlight the major pillars and enablers of Industry 4.0; cyber-physical systems, Internet of Things, Big Data, cloud computing, artificial intelligence, robotics, and Smart Factory.

### 1.1. Cyber-Physical Systems

The combination of physical and virtual spaces is referred to as *cyber-physical systems* (CPSs), and it aims to create a communicative interface between the digital and physical worlds by integrating computation, networking, and physical assets [5]. While the definition of CPS may vary based on perspectives and backgrounds, it is well-understood that the interconnection between the physical world is represented by hardware (e.g., sensors, actuators, robots) and cyber software (communication, networking and internet). CPS is at the core of Industry 4.0, and its success depends on the smart management of interconnected systems between its physical components and computational capabilities, utilizing state-of-the-art technology in both worlds [6].

CPSs have been an important topic in both research and the implementation of industrial technology ever since they were first introduced. The term Cyber-Physical Production System (CPPS) is used in the manufacturing industry to refer to the most recent advancements in computer science, information and communication technologies on the one hand, and manufacturing science and technology on the other [7]. CPPSs have been used in various industries in automotive, healthcare, and energy [8–11]. Further research on the sustainability of CPPS towards smart manufacturing, achieving a full life-cycle, and improving logistics, design, performance, and maintenance has been thoroughly discussed in [12] the references therein.

### 1.2. Internet of Things (IoT)

The International Telecommunication Union (ITU) defines IoT as *"A global infrastructure for the information society, enabling advanced services by interconnecting (physical and virtual) things based on existing and evolving interoperable information and communication technologies"* [13]. The infrastructure that allows devices and/or assets to connect to one another is known as the *IoT*. The term IoT has been deployed in many different fields, and it takes the name of the corresponding field; in Industrial internet of things (IIoT) [14], Internet of Service (IoS) [15], the merging of Robotics and IoT technologies gives birth to a new concept known as IoT-aided Robotics or Internet of Robotic Things (IoRT) [16].

In manufacturing systems, every component, including sensors, actuators, and electronic software, is embedded with physical devices, and it is all connected to various Internet networks. The Internet of Things makes it possible for manufacturing devices to share and exchange data both among themselves and with other manufacturing devices, as well as with the service providers and customers of manufacturing devices [17,18].

Recently, there have been numerous studies on IIoT and its uses in Industry and other applications. A number of studies offered insights and contributions that are focused on issues associated with protocols and standardizations [19], architecture and connectivity [20], and semantic and ontological issues for Industry 4.0 [21,22], while other studies focused on merging an IoT platform with CPSs towards smart digital manufacturing for additive manufacturing [23], in smart factory and smart manufacturing [24–26].

### 1.3. Big Data Analytics

Within the framework of Industry 4.0, Big Data Analytics has the potential to provide global feedback and a high degree of coordination, both of which are necessary to achieve high production efficiency. Data acquired from a variety of resources and channels, such as sensors, actuators, network traffic, and log files, can provide statistical findings for direct supervision and control duties, as well as dynamic reconfiguration and optimization of the system and enterprises, thereby empowering businesses [27].

The integration of Big Data Analytics and mining technology allows for the creation of intelligent analysis models and algorithms, which aid in the achievement of smart manufacturing. The digital twin paradigm of Industry 4.0 is one of the most prominent uses of Big Data [11]. Big-Data-driven manufacturing, which includes predictive manufacturing and proactive manufacturing, is another area where Big Data has made significant contributions [5].

### 1.4. Cloud Computing

Cloud computing can serve as a large platform for the company to store data, analyze information, and use upload applications to optimize their manufacturing chain, increase production efficiency, and reduce costs by providing a major hub for data exchange and enabling the sharing of a pool to store, process, and analyze data [28]. The industrial version of cloud computing is called *cloud manufacturing* (CMfg). CM has emerged with great impact due to its capability to integrate and utilize advanced technologies such as cloud computing, Big Data, CPS, and IoT platform [29]. As a result, it provides easy access to manufacturing resources and critical information, as well as virtualization and management of virtual resources, Big Data, and interfaces.

CMfg is distinguished by its flexibility and scalability, multi-tenancy, intelligent decision-making tool, and intelligent on-demand manufacturing [30] and has been adopted in a variety of applications, including IoT-enabled real-time machine status monitoring deployed in a CMfg environment [31], Visualization of RFID-enabled shopfloor logistics Big Data in Cloud Manufacturing [32], and cloud-computing for process monitoring and industrial control loops [33]. Further discussion and applications on CMfg can be found in [34].

### 1.5. Artificial Intelligence

Because of Industry 4.0, today's industrial environments are not only more responsive and connected than ever before, but they are also more complex than ever before due to rising levels of interdependencies, nonlinearity, uncertainty, and data volume [35]. The advancement of AI and the widespread use of Machine Learning (ML) and Deep Learning (DL)-based techniques across sectors are paving the way for these technologies to play a central role in the implementation of Industry 4.0. [36].

Regarded as one of the key technologies for Industry 4.0, AI focuses on the design, validation, and implementation of a wide range of machine learning algorithms for use in manufacturing and automation, which leads to industrial artificial intelligence [37]. AI-

based approaches have an impact on Industry 4.0 by enabling intelligent devices to conduct functions such as self-monitoring, interpretation, diagnosis, and analysis autonomously, resulting in increased agility, productivity, and sustainability.

There have been a number of industrial applications recently that have focused on how to incorporate and embed AI techniques and concepts within Industry 4.0. Some examples of these applications include Intelligent Data Analysis and Real-Time Supervision (IDARTS), which presents the guidelines for the implementation of scalable, flexible, and pluggable data analysis and real-time supervision systems for manufacturing environments [38], a complete framework and detailed overall planning of I-AI in the industry with an application scenario in [39]. The use of an artificial neural network for failure inspection in the assembly line of Industry 4.0 was proposed in [40]. In [41], real-time and online machine-learning approaches have been used for early failure identification in a real production environment for Industry 4.0. Other applications of AI in Industry 4.0 include predictive maintenance, sustainability, supply-chain management, and quality control (for an outlook, see [35]).

### 1.6. Autonomous Robotics

Robotics is one of the main pillars of Industry 4.0. Industrial manipulators, mobile robots, and collaborative robots are examples of robotics technologies involved in manufacturing processes and other Industry 4.0 applications. While industrial robots are considered the key technology in automation and industry, the latest technological innovations in the robotics field have boosted productivity, adaptability, versatility, reconfigurability, and safety in smart factories [42].

There are a variety of industrial applications that make use of cutting-edge robotics technology, such as using digital factory methods to create smart manufacturing cells [4,43], and in [44], a collaborative mobile robot has been integrated into an Industry 4.0 environment to achieve more flexible and intelligent production.

The merging of robotics and IoT technologies gives birth to a new concept known as IoT-aided Robotics or Internet of Robotic Things (IoRT) [16]. The fusion of robotics and IoT technologies will increase the capabilities of new creations in both the existing IoT and the robotics systems. This would facilitate the development of autonomous robotics and mobile robots that can be used in a variety of applications required for the advent of Industry 4.0 by creating adaptive and flexible systems. In [45], further reviews and applications of robotics within Industry 4.0 have been presented.

### 1.7. Smart Factory

The smart factory is an essential component of Industry 4.0, which focuses on achieving smart production through networked manufacturing systems and vertical integration of production processes [46]. The smart factory is a concept that makes intelligent use of robotics, automation, embedded systems, and information systems toward Industry 4.0. It is considered a transformation from classical (standard) to intelligent manufacturing heading towards digital manufacturing and digital twin models supported by many emerging technologies such as Cyber-Physical Systems (CPS), Internet of Things (IoT), Big Data, cloud computing, and advanced AI [17].

The autonomously running production processes of the smart factory are made possible by making full use of cutting-edge technology and manufacturing models to achieve self-optimized performance, adapt to new circumstances, and learn from them in real-time, enabling a high degree of flexibility, efficiency, and customization.

Several different fields of study have addressed the topic of "smart factory" in the published literature. In [47], an introduction to a smart factory architecture and its key enabling technologies and challenges, along with the application case, has been investigated. In [48], a systematic literature review on the smart factory was conducted in order to provide an overview of current research progress, advancements in this field, and promising future directions in this field. Another comprehensive literature review has been conducted in [49]

based on a multiple-stage approach and chosen criteria to establish existing knowledge in the field and to evaluate the latest trends and ideas of Industry 4.0 and smart manufacturing technologies, techniques, and applications in the manufacturing industry.

Some scholars focused on the key architecture of the smart factory and adopted a multi-layer description consisting of four main layers [46,47]:

1.  Physical resources layer: it includes all manufacturing resources (machines, equipment, sensors, actuators).
2.  Network layer: data transmission and sharing between different layers (mainly between the physical resource layer and the cloud layer) requires advanced network technologies and communication protocols for high-speed and reliable real-time communication. It utilizes advanced networking technologies and protocols such as industrial ethernet, industrial wireless networks, Fieldbus, Profibus, Wi-Fi, Bluetooth, CAN, Cellular 4G/5G, TCP, OPC-UA, MQTT, JSON.
3.  Data Application layer: also called *cloud layer,* where the term *cloud manufacturing (CMfg)* has recently been widely used within smart factory context [24]. This layer serves as an information hub center for the smart factory, where cloud computing technology enables various Big Data applications, including data exchange, data storage and data analysis, and data management [50].
4.  Terminal layer: In this layer, people are connected to the smart factory assets and resources and available information via end-user devices (terminals), including smartphones and tablets, PCs, and monitoring devices. This provides human intervention via remote supervision to the processes and visualization of the results from CMfg for real-time diagnosis and control of the factory or prediction for future actions based on feedback performance indicators.

Figure 2 shows the architecture of the smart factory where the four layers are integrated via different levels of an ***intelligent*** *automation pyramid*. This pyramid is different from the classical one as it enables information exchange and interaction between the elements of the hierarchical levels of the industrial process via advanced computing, clouding, networking, and smart devices.

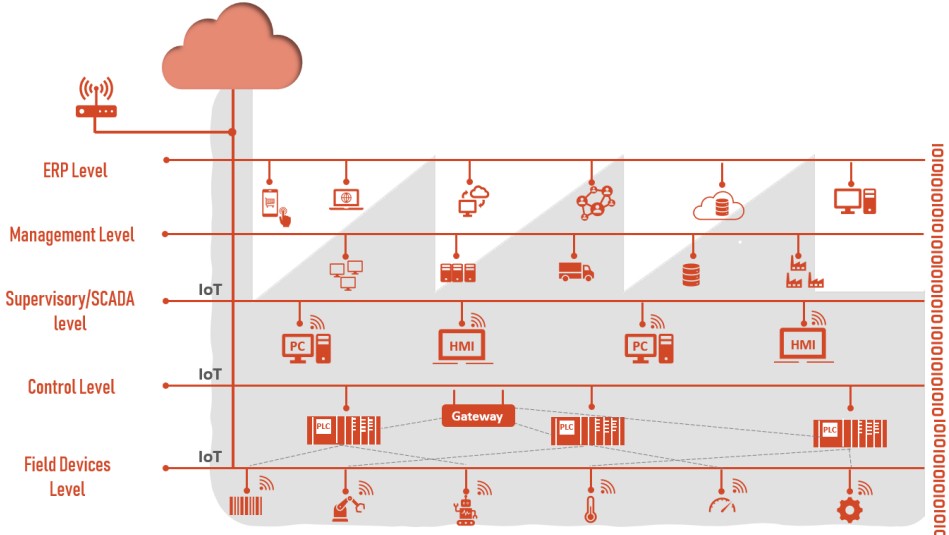

**Figure 2.** The architecture of a smart factory in an intelligent automation pyramid.

Within the Industry 4.0 framework, smart factories have been applied to a variety of industrial enterprises, including a smart factory for drug packing in pharmaceutical production of the Healthcare Industry 4.0 [51], a framework and operational mechanism of a smart factory utilizing an intelligent negotiation mechanism to enable self-organization of agents [27], the use of digital twin technology for a CPS-based smart factory for circuit breaker manufacturing in [52], the application of a smart factory in additive manufacturing

in [53], real-time optimization of a standard chemical plant towards a smart plant according to the Industry 4.0 paradigm in [54], and deployment of Industrial IoT towards conversion from the standard factory to a smart factory in the tannery industry [55]. Some works focused on smart manufacturing and special machining, including a method proposed in [56] for reducing machining problems by utilizing a simulation utility that uses the primary system and processes variables as input data and generates results (intelligent utility) that aid in decision-making and machining planning, and a systematic framework for smart manufacturing systems in Industry 4.0 was proposed in [57], which covers a wide range of topics, including design, machining, monitoring, control, and scheduling. Reference [58] provides more applications and future perspectives of Smart Factory.

*1.8. A Note on Industry 5.0*

Industry 4.0 has been the dominant theme in industrial trade for the past decade. It has become a globally accepted paradigm with a profound impact on economic and social development. It is characterized by its intelligent and autonomous production systems. Such an automation-centered viewpoint has been called into question in recent years, leading to the proposal of an Industry 5.0 concept, which places the worker at the center of digital transformation [59]. Among the published literature, five major themes of Industry 5.0 were identified: supply chain evaluation and optimization, enterprise innovation and digitization, smart and sustainable manufacturing, IoT, AI, Big Data, and human-machine connectivity. Industry 5.0 as a gateway to human–machine interaction and coexistence is particularly popular among researchers [60]. It aims at redefining the relationship between humans and intelligent components toward a collaborative industrial system with comprehensive human-centered development, a long-term contribution to sustainability, and a reliable source of resilience [61].

This paper presents the design of a smart cyber-physical system that conforms to the novel smart factory framework towards Industry 4.0 and deploys the key technologies of the smart factory, including industrial, computing, information, and communication technologies. It describes the integration of the basic layers of the smart factory for the creation of an intelligent manufacturing system. A case study of smart manufacturing, including a drilling process, is implemented as a demonstration for a simplified smart factory model.

The remainder of this paper is organized as follows. Section 2 describes the methodology and overall architecture of the proposed smart factory. The implementation and validation of the proposed model, along with a discussion of the results, are presented in Section 3. Section 4 gives the conclusions, and finally, Section 5 presents the recommendation for future work and development of a smart factory in Industry 4.0.

## 2. Methodology and System Architecture

This section discusses the proposed smart cyber-physical system consisting of the layers mentioned in Section 1; the physical resources layer of the smart factory platform is shown in Figure 3, which is composed of a robotic arm (KUKA KR 6 sixx r900), PLC Siemens S7-1200, universal web camera, variety of sensors, a drilling unit and a PC with a range of communication technologies to be discussed later.

The considered smart manufacturing is a machining (drilling) process where the workpiece is transported to a certain slot where the PLC and the web camera detect its arrival. The machine vision system calculates the position and orientation of the workpiece in order for KUKA robot to know the coordinates of the workpiece's gripping point. The robot that is responsible for pick-and-place tasks receives the corresponding position and picks the item and places it in the designated drilling unit slot. A proximity sensor is used to check its arrival at the drilling table to start the drilling process. A limit switch is triggered after the drilling process has been completed so the robot can move the workpiece to the designated storage unit. This process repeats itself each time a piece is present at the starting slot. The PLC is programmed using the ladder logic via a *Totally Integrated*

*Automation (TIA)* portal, and the KUKA is programmed using KUKA Robot Language (KRL) via its own Human–Machine Interface (HMI).

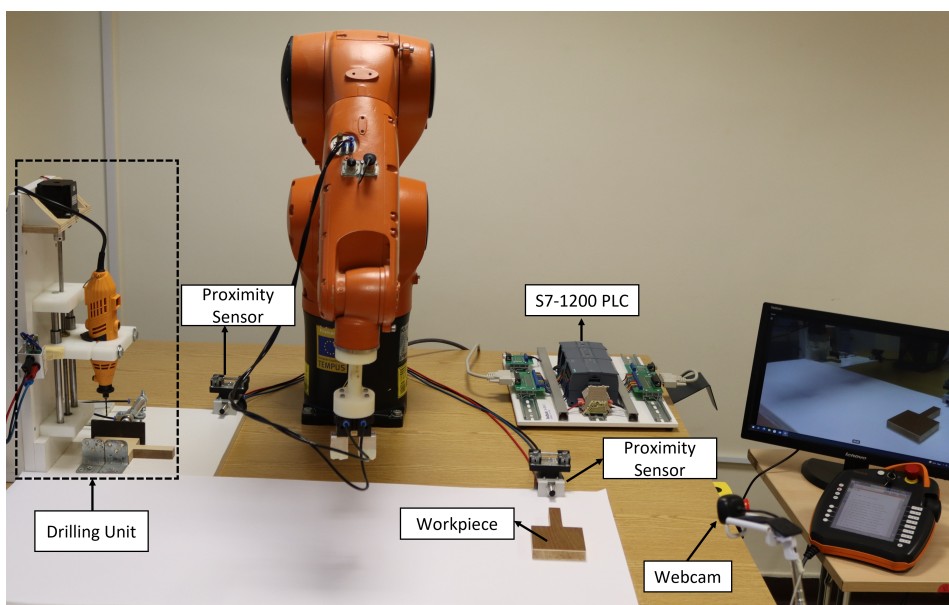

**Figure 3.** Experimental platform of the proposed smart factory.

The sensors used are connected to the PLC, and the LabVIEW Application Program Interface (API) transfers data between the KUKA robot and the PLC to the cloud. The following subsections describe the system architecture in more detail.

### 2.1. The Mechanical Design

In this work, a vertical drill was designed to demonstrate a mini-factory process, and the KUKA was programmed to pick the specimen from a designated slot and transport it to the drilling platform to be drilled. A drilling machine is a device for producing holes in workpieces.

The CAD model of the drilling platform (unit) was proposed, as depicted in Figure 4. Then, a real system was constructed based on some engineering calculations, as depicted in Figure 3. The drilling system consists of the following components:

1. Mechanical components: includes the structure of the drilling machine, i.e., the base, support structure, beams, lead screw, and bearings.
2. Electrical system: the electrical system consists of a stepper motor and its driving power and control units.
3. Mechanism control: it positions the tip of the drill at the required position and provides the depth of cut.

A gripping mechanism was attached to the drilling unit, as shown in Figure 3, to fix the workpiece and ensure stable operation during the drilling process. This mechanism includes having a pneumatic piston and a metal gripper (clamp) installed on the table base so that the workpiece is fixed from both sides.

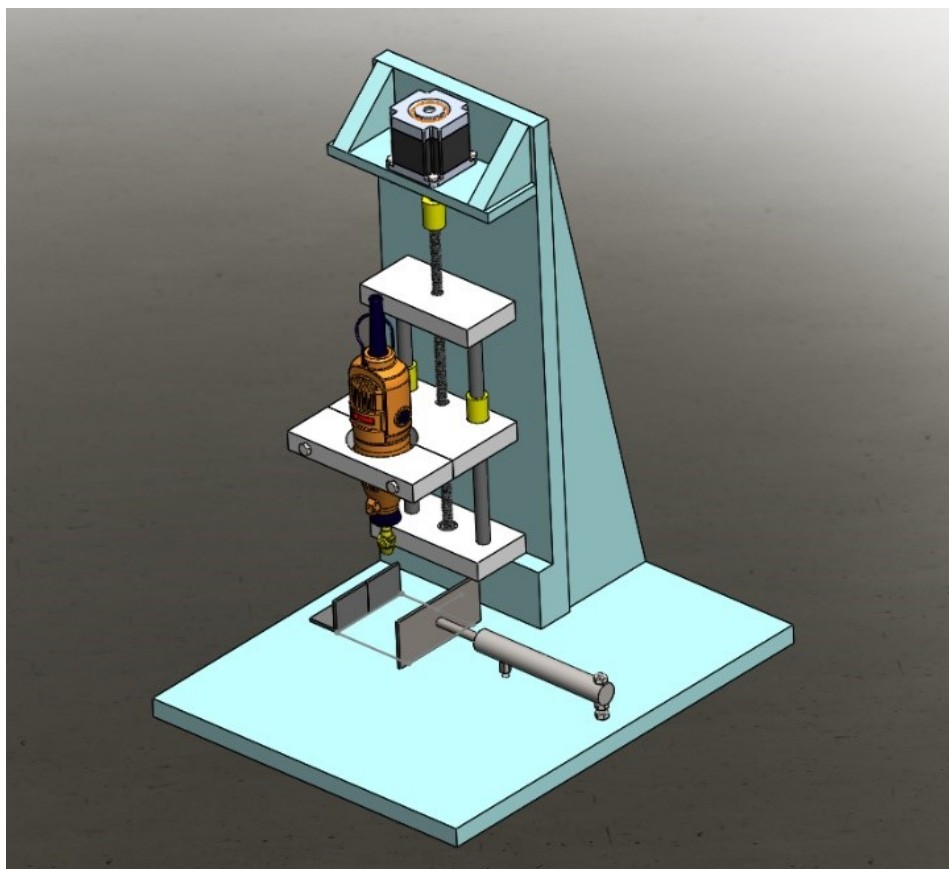

**Figure 4.** The proposed CAD model of the drilling unit.

### 2.2. Process Control

Sequential actuation of the mechanism was carried out using an Arduino microcontroller, which communicates with the PLC (the main controller). Three sensors are connected to the PLC; two proximity sensors were used, the first one detects the arrival of the workpiece to initiate the process, and the second sensor detects the arrival of the workpiece to the drilling unit. The third sensor was a switch-activated sensor that is triggered by the linear vertical motion of the drill. The second proximity sensor signal (checks the arrival of the workpiece to the drilling table) is fed to the Arduino through the PLC.

The PLC outputs only 24/0 volts as digital outputs, whereas the Arduino operates on a 5/0 logic level. Therefore, an industrial optocoupler is used to convert the 24/0 digital logic to 5/0 digital logic. This signal initiates the drilling process where the Arduino controls a stepper motor that moves a table vertically to the designated position. The workpiece is also clamped by the pneumatic cylinder to hold it in place. The mechanical drill and the pneumatic cylinder are actuated by relays withdrawing signals from the Arduino microcontroller.

### 2.3. IoT Platform Configuration

The IBM Watson IoT Platform is a hub for connecting devices, gateways, and applications for IoT solutions. REST and MQTT protocols are supported within this platform for applications, devices, gateways, event processing, and administrative tasks. This platform bridges the gap between field devices and data analytic services of IBM services or online user applications. The easy interfacing through the MQTT protocol as well as the user-friendly platform sufficed for this application.

In the *Devices* tab shown in Figure 5, devices can be added and managed. Each device that connects to the IoT platform must be configured with a device ID, type, and authentication token. External devices are connected to the platform according to their parameters.

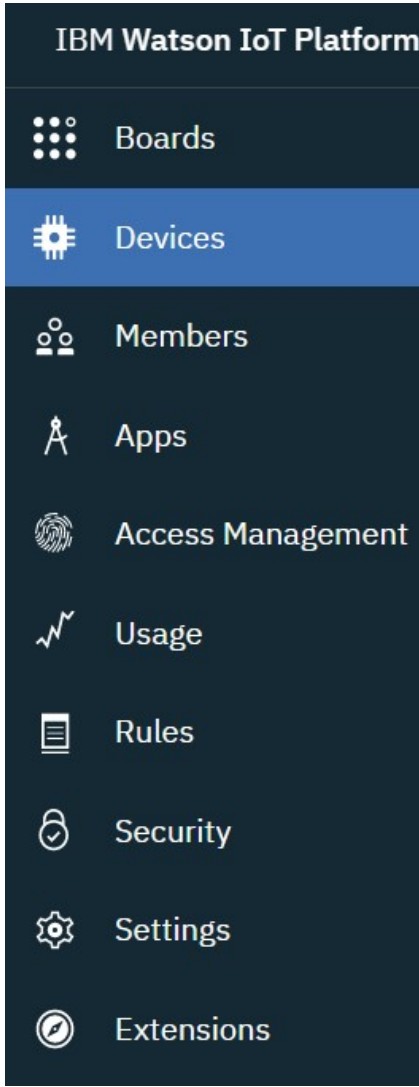

**Figure 5.** IBM Watson menu.

In the *Boards* tab, real-time data can be visualized by creating cards with many different visualization options. In the *Apps* tab, Application Programming Interface (API) keys can be added and generated for different user applications that can connect to the IoT platform and subscribe to devices' topics or publish commands to devices.

In this paper, the following connection information for the MQTT client was used:

- Messaging Address: uve180.messaging.internetofthings.ibmcloud.com, where uve180 is the name of the organization
- Security Port: connection type non-secure, protocol MQTT, port number 1883;
- MQTT Client: device;
- Device Authentication: token authentication;
- QoS: at most once (QoS0).

The data are sent to the platform along with a topic and a property describing the data.

### 2.4. Communication and Interfacing

In the course of the implementation within the CPSs context and to ensure hardware compatibility with Industry 4.0, different system extension approaches were investigated analogous to the extension by a microcontroller board that integrates the PLC S7-1200 with the KRC4 through LabVIEW API, enabling communication to-and-from both manufacturing hardware, as well as cloud-devices interactions. Figure 6 illustrates the system interaction.

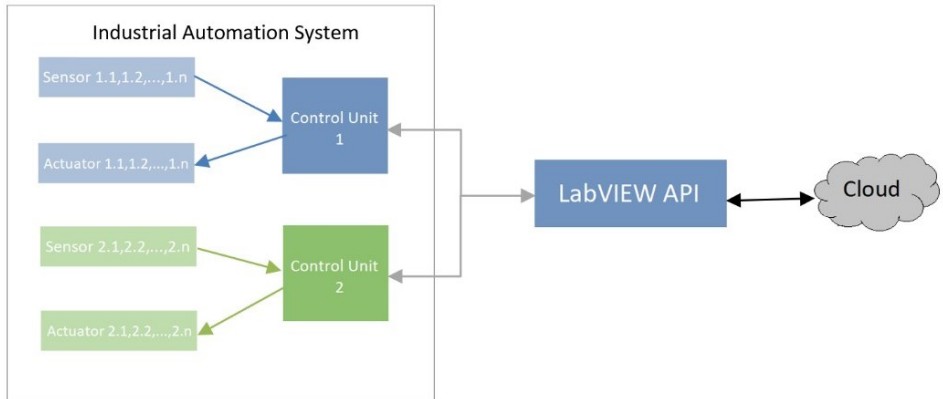

**Figure 6.** System interface using microcontroller.

The challenge behind the network is the ability to communicate with both the PLC and the KUKA, without using their respective accessories, in other words, connecting the PLC using its S7 protocol to the KRC4 through external applications. Currently, there is no standard protocol released by KUKA and Siemens available for mutual usage (i.e., outside their specific environment). Communications between devices are established using a network protocol analyzer *Wireshark*, and then a library is written to read/write PLC data blocks, KRC4 variables, and data structures by using the observed protocol.

### 2.5. The PLC Module

In this application, Siemens S7-1200 was adopted and communication with the S7-1200 was first approached using the TCP functional blocks in *STEP 7* development environment from Siemens. In the ladder program, these functional blocks are used to initiate TCP/IP communication with other ethernet capable devices.

Within this framework, the S7 protocol is the one which Siemens use to exchange data between PLCs, in addition to accessing the PLCs from SCADA systems. It implements the ISO-on-TCP, which is a block-oriented protocol that fulfills the TCP/IP standard. The S7 protocol is function/command-oriented, which means each communication is a command or a reply to a command. The command of interest is Data Read/Write. This interfacing option does not require TCP socket programming in the ladder program. Wrappers for the Snap7 are available in C, C++, and Python.

Fortunately, the S7 protocol has a LabVIEW VI library allowing reading and writing to data blocks. This was adopted but with some modifications to facilitate the objective of this process.

### 2.6. Communications with KUKA Robot

JOpenShowVar is an open-source java cross-platform communication interface to the KUKA robot via its KRC4 controller. Along with its complementary KUKAVARPROXY that runs on the KRC4, they can be used to read/write global variables of the KRC4. The externally controllable and observable global variables can be used in a KRL program whose flow can be altered based on them. This interface is established by connecting the KRC4 via ethernet from the ethernet safety interface (X66) to a PC.

The reading and writing of variables and data structures of the controlled KuKa manipulators are facilitated through "JOpenShowVar", which is a Java open-source cross-platform communication interface with KUKA industrial robots. Figure 7a shows the architecture of the application program interface (API) for 'JOpenShowVar', which is a client that runs on a remote PC and interacts with user applications. The interface and the interaction with the real-time control process of the manipulator are established through KukavarProxy, which is a TCP/IP server that enables the reading and writing of robot variables over the network. It runs on the KRC4 and allows up to 10 clients to connect. The class responsible for the communications is the "CrossComClient", whose API is shown

in Figure 7b. The functions 'writeVariable' and 'readVariable' are the ones responsible for reading from and writing to global variables. This class takes the server (KRC4) IP address as well as the corresponding port.

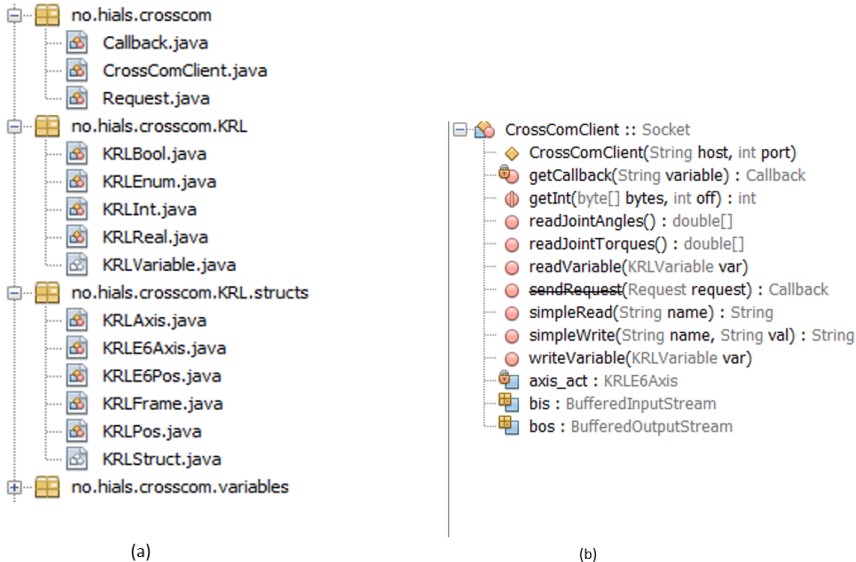

(a)　　　　　　　　　　　　　　　　　(b)

**Figure 7.** (**a**). The API architecture of JOpenShowVar; (**b**). The API architecture of CrossComClient.

KRL and KRL.structs are classes that represent actual variables from different data types in the KRL. This class takes in a string name that represents the exact name of the global variable defined in the KRC4.

*2.7. LabVIEW API*

LabVIEW API follows the classical producer/consumer design pattern. This design pattern is used to process data being produced and consumed at different rates to avoid data loss. For example, the data exchange between the JOpenShowVar API is completed "at a non-deterministic time with an average of 5 ms". Since the process of uploading the data to the IoT platform is dependent on the speed of the internet connection, it could take hundreds of milliseconds to a few seconds. Figure 8 depicts an example of two loops running asynchronously. The producer loop actively polls the variables from both the PLC and the KRC.

The producer loop shown in Figure 9 is a state machine with eight states:

- Initialize: initializes TCP client sockets.
- Test_Internet: tests connection to the internet and its stability.
- Connect_IoT: connects devices to the IBM Watson IoT platform.
- Check_Status: checks the termination of the program.
- From_PLC: reads a data block from the S7 1200 and enqueues read variables.
- From_Kuka: reads from the KRC as well as enqueues the read variables.
- To_Kuka: writes to the KRC.
- Stop: stops program execution.

In both From_PLC and From_Kuka states, the strings read are parsed and formatted according to a specific protocol. The consumer loop does not run until there are data in the queue. The data are then dequeued and parsed to be published to the IoT platform with their respective property and topic.

Although the MQTT protocol runs over the TCP/IP, it is better to use a library that handles the details of the MQTT protocol. The Espotel IoT foundation library is chosen for IoT communications. The library eases the connection with IBM Watson IoT using MQTT. It contains blocks for connecting devices and applications, as well as publishing events and subscribing to events.

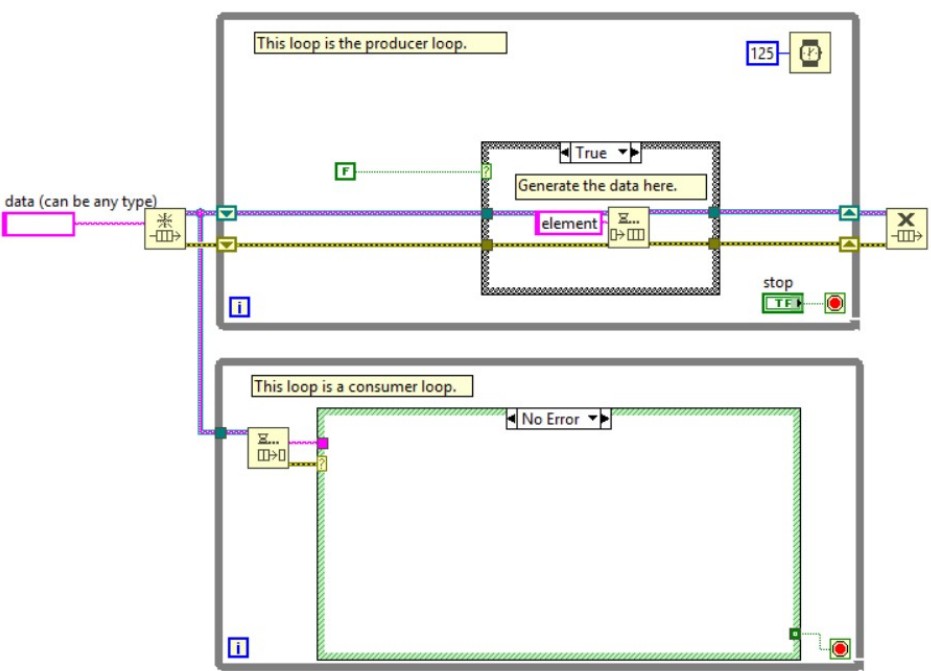

**Figure 8.** Producer–consumer loops.

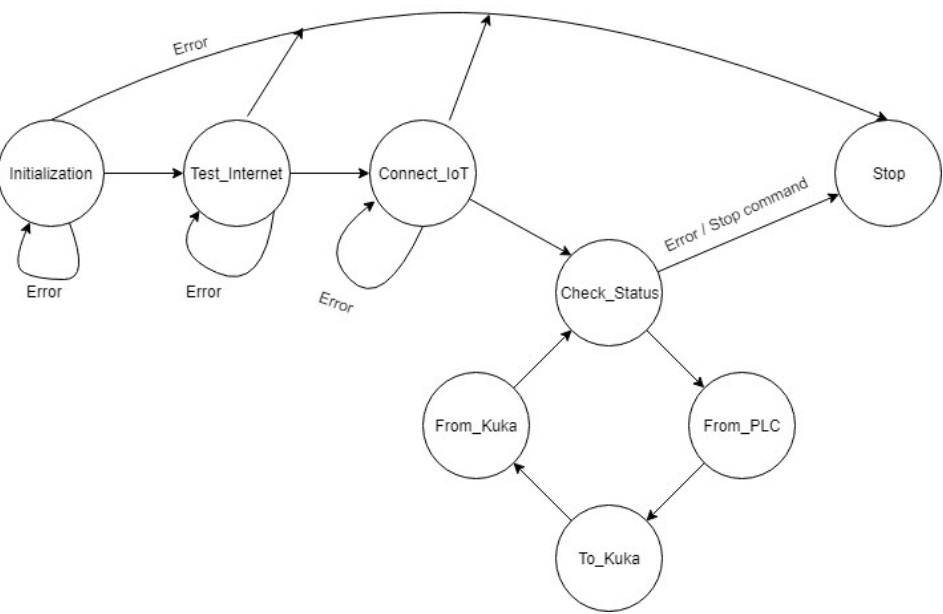

**Figure 9.** Producer loop state diagram.

### 3. Implementation and Discussion

The smart factory application is a drilling process that showcases the integration of a KUKA robot (KR 6 sixx r900) with Siemens S7 PLC. A TCP network is built between them to facilitate communication.

The process starts by placing a workpiece on a defined slot opposite to an ultrasonic sensor. The position of the workpiece is transmitted from the sensor and camera via the controller to the robot. The robot then moves to pick up the workpiece and places it on the drilling table. After placing the workpiece, the controller actuates a pneumatic cylinder that holds the workpiece in place and starts a lead screw mechanism which has a drill. The drill moves down vertically and drills the workpiece after the drilling is complete.

The robot picks up the workpiece and places it in the designated stack and waits for another workpiece.

Figure 10 shows the interfaces between the main components of the system, and Figure 11 shows the detailed program data flow.

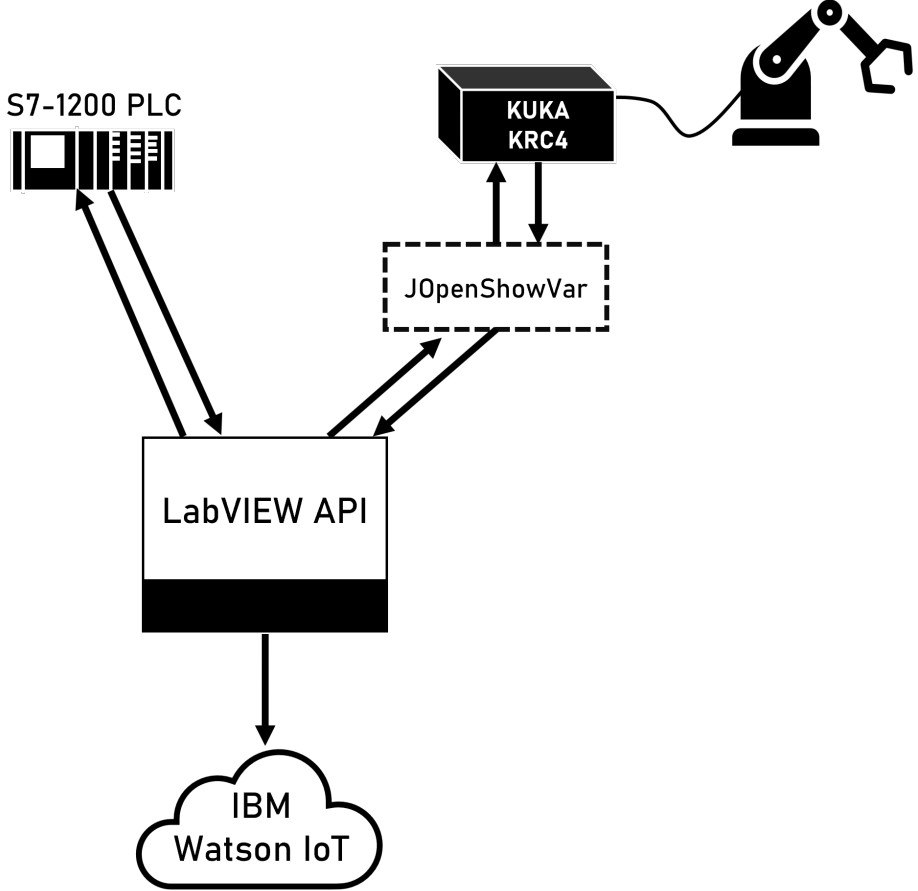

**Figure 10.** Interfacing network.

The LabVIEW API creates two client sockets for both the PLC and the JOpenShowVar server program. In the API's producer loop, the incoming data from the two devices are parsed, encapsulated, and enqueued. In the API's consumer loop, the data are dequeued, decapsulated, and published to the IoT platform.

Variables of interest from the KUKA robot:

1. $AXIS\_ACT\_MEAS$: returns the axis angles of the current robot position.
2. $VEL\_AXIS\_ACT[axisnumber]$: returns the current speed as a percentage of the maximum speed of the axis number.
3. $MOT\_TEMP[axisnumber]$: returns the current speed of the motor of the axis number as a percentage of maximum speed.

Variables of interest from S7 1200:

1. $Dint\_12$: counts the number of workpieces passed by the conveyor belt.
2. Boolean array of 8 bits that is used to indicate the position of the workpiece based on the sensor readings.

Two java projects are created: The first is JOpenShowVar-core-client1, which is responsible for exchanging the actual process parameters with myRIO. Its main function is in Newproject.java, and four global variables are defined to be exchanged with myRIO. These variables are defined and configured in KRC. The KRL program is structured in a way that allows these exchangeable global variables to control its flow.

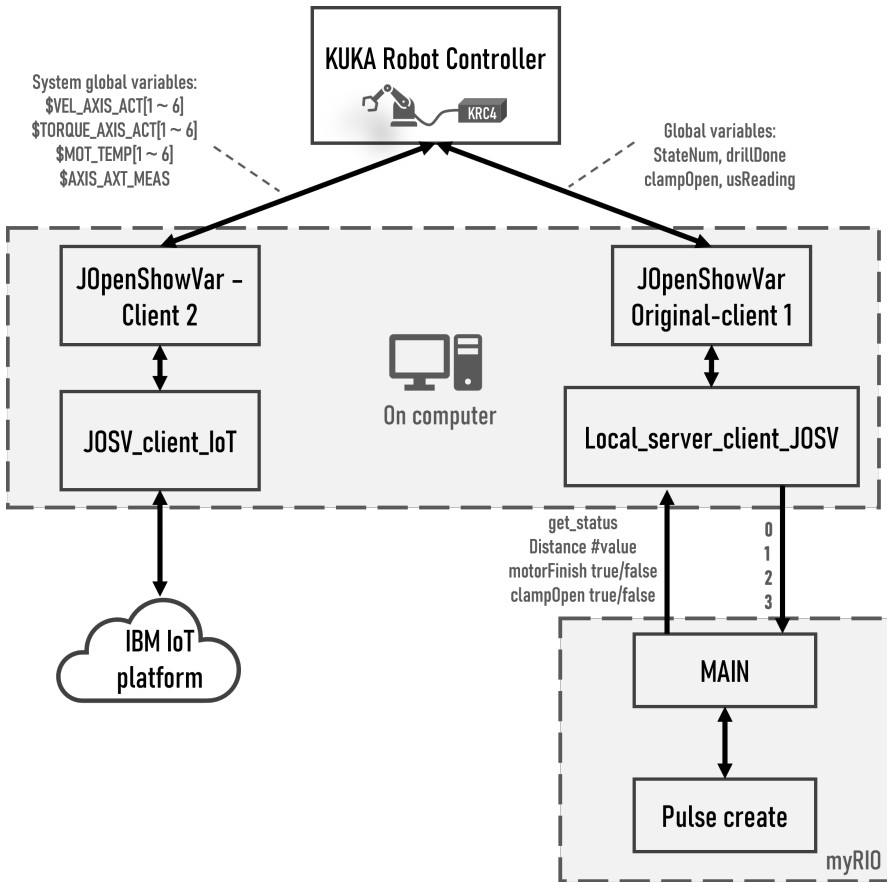

**Figure 11.** Program data flow.

The four variables are also defined in NewProject.java. The program enters a while loop and reads the TCP socket input stream. It reads one line at a time. The read string is then split by whitespaces. If the length of the string array after the split is zero, then that means that a read operation is required. The split elements are then compared with $get_s tatus$, which gets the current status of the robot. In the application, four states are distinguished:

- Status 0 indicates the robot is busy with a motion block.
- Status 1 indicates that the robot is waiting for a valid distance to the workpiece to pick it up and moves it to the drill table.
- Status 2 indicates that the robot is waiting for the drilling process to be finished.
- Status 3 indicates that the robot has clamped the workpiece and is waiting for the pneumatic release of the cylinder

If the length of the string array is 2, then that means that a write operation to one of the variables has to occur. The first string indicates what variable to write to and the second is the value to be written.

The second is $JOpenShowVar - core - client2$, which is responsible for sending the robot's sensor readings such as velocity, torque, axis motors' currents, and axis motors' temperature to the IBM Watson IoT platform through $JOSV_c lient_IoT$. This data can be accessed from the system variables[see KUKA manual system variables], which are read-only global variables.

Note that in the KRL program, there is WAIT SEC 0. These lines serve the purpose of stopping the advance run pointer. The advance run is the maximum number of motion blocks that the robot controller calculates and plans in advance during program execution. This can set values to variables before the program finishes motion.

### 3.1. Response Time of the System (PLC-KUKA Interface)

Table 1 describes the calculation of the response time for the interface between the PLC and the robot. The table shows the average responses for several experiments. The producer loop, which is responsible for the data exchange between the devices has an iteration speed of about 100 ms, which is a good response time.

**Table 1.** Time required for a data block transmission within the local interfacing network.

| Variable | Time (ms) |
|---|---|
| Average Time from KRC4 to LabVIEW API | 8.744 |
| Average Read Time from PLC Data Block | 2.537 |
| Average Producer Loop Iteration Time | 104.3 |

### 3.2. System Integration with IoT

Table 2 illustrates the average time for an MQTT message to reach the IoT platform. The JSON string is used in the speed test. It can be noted that the QoS0 level of the MQTT protocol needed the least time to transfer the data to the cloud; therefore, it was chosen for the application. On the other hand, the QoS2 level is needed the most since increasing the number of iterations will increase the time needed for data transmission.

**Table 2.** Time required for an MQTT-IoT message transmission.

| Variable | Time (ms) |
|---|---|
| Average connection time | 264.3 |
| Average publish time (QoS0: at most once) | 0.9215 |
| Average publish time (QoS1: at least once) | 108.4 |
| Average publish time (QoS2: exactly once) | 217.5 |

Figure 12 shows the visualization of some measured parameters in IBM Watson IoT. Once the data reach the IoT platform, services such as Watson Analytics or IBM Data Science Experience can be used. Storage of data is also possible with the Cloudant NoSQL database. IBM Watson also allows the binding of other user applications with the IoT platform.

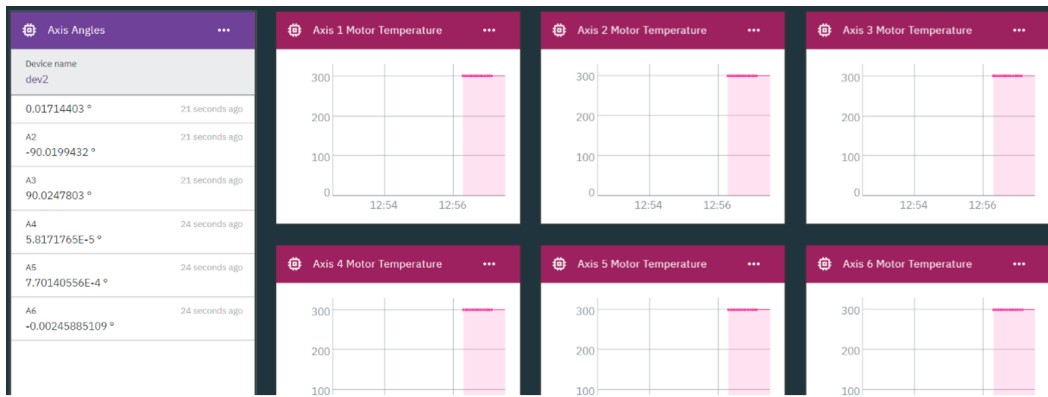

**Figure 12.** Data visualization in IBM Watson IoT.

## 4. Conclusions

Industry 4.0 has emerged as a primary focus in the manufacturing sector in response to the rise of digitalization and smart manufacturing. In the era of Industry 4.0, a smart factory represents a core pillar in digital transformation. It paves the way for a smooth transition from classical (standard) to intelligent production with the help of cutting-edge innovations in intelligent robotics, cyber-physical systems, smart sensors, and instrumentation, the internet of things, cloud computing, Big Data, and artificial intelligence.

In this paper, a hierarchical design for a smart factory that takes into account the most recent findings in the field is proposed. This design, which consists of all of the advanced technologies associated with all smart factory layers, has been implemented and verified in a test-bed platform, which has shown the effectiveness of the adoption of the smart factory in manufacturing processes.

With this method, it can be seen how the adoption of the smart factory paradigm with its high-tech components has led to a more efficient manufacturing process, better data exchange, and a greater degree of autonomy. The findings revealed that the framework promotes communication and cooperation among system modules, resulting in a faster production process (decreased response time) and high productivity. One of the limitations of the proposed method is that the model's application scope is limited to a single production theme. The smart factory aims to produce products that are more adaptable and customizable through reconfigurable manufacturing. This subject, along with those mentioned in Section 5, will be further investigated in future works.

## 5. Future Directions

Improvement in any smart factory can be achieved by including more enablers/pillars of Industry 4.0, which will be discussed in the following subsections. Further improvements can also be achieved by the selection of advanced microcontrollers and embedded systems that can be used as interfacing options for enhanced communication speeds between the devices. Adding "smartness" to the smart factory can also be obtained by deploying a *digital twin*. Digital twins are cyber digital models of physical components, products, or systems that can provide a comprehensive physical and functional description in a *mirrored* digital environment for the purpose of simulating their behaviors in real-world environments [49]. By creating virtual models synchronized with the physical models, the simulation of various scenarios contributes to the prediction and prevention, as well as optimization of the manufacturing process [62].

### 5.1. Cybersecurity

A big challenge to the implementation of the smart factory is the protection of the system from cybersecurity threats. Industry 4.0 is characterized by increased adoption of IoT technologies and cloud computing and increasingly relies on Big Data and advanced ICT infrastructure. This increased dependency will also lead to increasing new vulnerabilities to *cyberattacks* and privacy risks.

Smart factory security can be investigated based on an in-depth literature review in order to explore the current cybersecurity approaches and trends, the challenges associated with the integration of cybersecurity in the smart factory architecture, frameworks, and standards, the types of attacks and threats, and their risk mitigation on different levels (device, data, and individuals' privacy) [63–67].

### 5.2. AI

As discussed in Section 1.5, artificial intelligence is a core and important enabler of Industry 4.0. With the advancement of Big Data technology, data-driven modeling from AI (machine and deep learning) will improve product quality and increase production efficiency via self-optimization and predictive actions. Furthermore, artificial intelligence can be deployed on the cloud-based database after storing the IoT data generated, which has provided the availability of statistical data extracted by the system and subsequently the ability to transmit the analyzed data back to the system for decision-making, such as diagnosis, predictive maintenance, reconfigurable manufacturing, and optimized manufacturing.

Various methods can be used, starting with the built-in machine learning toolbox of IBM Watson. Artificial Neural Network (ANN), support vector machine, random forest, and Bayesian networks are all examples of common machine learning methods that have been demonstrated in a variety of industrial contexts [68–70].

**Author Contributions:** Conceptualization, M.R., H.E. and M.A.; Methodology, M.R., H.E. and M.A.; Software, M.R., H.E. and M.A.; Validation, M.R., H.E. and M.A.; Formal analysis, M.R., H.E. and M.A.; Investigation, M.R., H.E. and M.A.; Resources, M.R., H.E. and M.A.; Data curation, M.R., H.E. and M.A.; Writing—original draft, M.R., H.E. and M.A.; Writing—review & editing, M.R., H.E. and M.A. All authors have read and agreed to the published version of the manuscript.

**Funding:** This research received no external funding.

**Institutional Review Board Statement:** Not applicable.

**Informed Consent Statement:** Not applicable.

**Data Availability Statement:** Not applicable.

**Conflicts of Interest:** The authors declare no conflict of interest.

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
