# Peer review of "Design of a Smart Factory Based on Cyber-Physical Systems and Internet of Things towards Industry 4.0"

_applsci, doi:10.3390/app13042156_

Round 1
Reviewer 1 Report
Minor concerns are:
1. the manuscript presents a very interesting case study of a smart cyber-physical system that conforms to the Smart Factory framework towards industry 4.0 and deploys the main key enabling technologies. The basic layers of the smart factory are indeed well integrated to the creation of an intelligent manufacturing system
2. the topic is relevant (not original since concepts of industry 4.0 is nowadays very well spread)
3. the conclusions are consistent and pertinent
4. References are wide and all appropriate; anyway, authors are suggested considering the following three works:
1) Optimally Managing Chemical Plant Operations: An Example Oriented by Industry 4.0 Paradigms, Industrial and Engineering Chemistry Research, 2021, 60(21), pp. 7853–7867,
2) An automatic system for modeling and controlling color quality of dyed leathers in tanneries, IFAC-PapersOnLine, 2021, 54(3), pp. 164–169
as two examples of Industrial IoT towards a conversion from standard factory to smart factory.
3) A cloud-based monitoring system for performance assessment of industrial plants, Industrial and Engineering Chemistry Research, 2021, 59(6), pp. 2341–2352; as an example of cloud-computing for process monitoring and industrial control loops.
5. Figure 5 is not very significant (being included in Figure 5) and could be saved
6. a general revision of the English is suggested; some minor typos are present along the text. For example: line 75 “big”; line 225: “a proximity sensor”; line 372: a coma not a full stop.
7. the paper is very tutorial, easy to read and understand.. it deserves publication
Reviewer 2 Report
The authors have done valuable and important work.
The Manuscript proposes the design of a smart cyber-physical system which is interesting and well described.
The selected technologies and methodologies are well integrated and the developed architecture of the proposed kind of a smart factory as well as the CPS are well considered and developmental.
The proposed implementation of the experimental platform of the proposed smart factory and its validation (including models) are clear, comprehensive and interesting. The results and conclusions are sufficient.
There can be a following suggestion considering Fig. 1: the authors can consider including the following:
- "Manufacturing" splits into (includes) the elements: "Automation and Industrial Robots". "Cyber-Physical Systems", and "Additive Manufacturing".
- "Integration" splits into (includes) the elements: "Simulation and Modelling", "Blockchain", and "Augmented Reality".
Please check and correct where appropriate:
"1.3. Big data Analytics" (line 74) instead of "1.3. Big data Analytic"
"big data analytics" (line 81) instead of "big data analytic"
"paradigm of Industry 4.0" (line 83) instead of "paradigm of industry 4.0"
"Industry 4.0." (line 129) instead of "industry 4.0."
"Big Data" (line 94) instead of "big data"
"and IoT platform" (line 94) instead of "and Iot platform"
"Industry 4.0." (line 129) instead of "industry 4.0."
"Internet of Robotic Things (IoRT)" (line 140) instead of "Internet of Robotic things (IoRT)"
"communication. It utilises" (line 175) instead of "communication. it utilises"
"including a drilling process" (line 207) instead of "including drilling process"
"in Section 3. Section 4" (line 211-212) instead of "in Section 3 .Section 4"
"where the PLC and the web camera detect" (line 221) instead of "where the PLC and the web camera detects"
"for reading from and writing to" (line 331) instead of "for read from and writing to"
"This class takes in" (line 331) instead of "This class take in"
"Its main function is in Newproject.java and 4 global variables are defined" (line 392-393) instead of "Its main is Newproject.java 4 global variables are defined"
"decision making such as diagnosis" (line 475) instead of "decision making such diagnosis"
"Machine Learning methods which have been" (line 479) instead of "Machine Learning methods have been"
The sentence from line 194 to line 200 has syntax, punctuation, and grammar issues, for example in "Healthcare Industry 4.0 [49], A framework" (should be "." instead of ","? or else should be ", a framework"), "...agents has been proposed in...", "...using digital twin technology a CPS-based smart factory..." (should be "and" instead of "a" ?), ...
The Manuscript can be recommended for publication in the journal.
Reviewer 3 Report
The paper “Design of a Smart Factory based on Cyber-physical Systems and Internet of Things towards Industry 4.0” presents an approach for developing an intelligent manufacturing system, adopting the smart factory paradigm in industry. From my point of view, the article is interesting, but some comments:
· In the abstract when the author mention the novelty of the paper it is not clear explanation.
· Nice introduction of the topic, this industry 4.0 has gather a lot of developments under this umbrella and it is not an easy task to summarise. Nevertheless, I miss some mention to the upcoming industry 5.0.
· In the intro, cite some other papers more focus on manufacturing and in special machining:
o https://doi.org/10.3390/machines5020015
o https://doi.org/10.1007/s11465-018-0499-5
· Figure 8 caption is not adequate Please improve it.
· Figure 11 is in the middle of a paragraph.
· Same on Figure 12
· A lot of typo errors (as an example “compnents” line 375)
· Improve quality of Figure 13
· Avoid the use of personal verb forms, like: “we proposed a hierarchical design for a smart factory that”
· The conclusions have to be reworked, right now it is a summary of what has been done. Advantages, disadvantages, significant quantifiable results... are expected.
· Personally, I do appreciate the inclusion of a Future work section.
Finally, I appreciate the amount of work done and the importance of projects like these. I do not consider that in the theoretical part it is a great contribution to the subject, but in the practical part it is outstanding for its application at scale.
Round 2
Reviewer 3 Report
It has improved